

# The bony cap and its distinction from the distal phalanx in humans, cats, and horses

Shannon Smith[1], Laurel R. Yohe[2,3,4,5] and Nikos Solounias[1,6]

[1] College of Osteopathic Medicine, New York Institute of Technology, Old Westbury, New York, United States
[2] Bioinformatics and Genomics, University of North Carolina at Charlotte, Charlotte, North Carolina, United States
[3] Earth and Planetary Sciences, Yale University, New Haven, Connecticut, United States
[4] North Carolina Research Center, Kannapolis, North Carolina, United States
[5] Ecology and Evolution, State University of New York at Stony Brook, Stony Brook, New York, United States
[6] Department of Paleontology, American Museum of Natural History, New York, NY, United States

Corresponding authors
Laurel R. Yohe, lyohe1@uncc.edu
Nikos Solounias, nsolouni@nyit.edu

## ABSTRACT

It has been recognized as early as the Victorian era that the apex of the distal phalanx has a distinct embryological development from the main shaft of the distal phalanx. Recent studies in regenerative medicine have placed an emphasis on the role of the apex of the distal phalanx in bone regrowth. Despite knowledge about the unique aspects of the distal phalanx, all phalanges are often treated as equivalent. Our morphological study reiterates and highlights the special anatomical and embryological properties of the apex of the distal phalanx, and names the apex "the bony cap" to distinguish it. We posit that the distal phalanx shaft is endochondral, while the bony cap is intramembranous and derived from the ectodermal wall. During development, the bony cap may be a separate structure that will fuse to the endochondral distal phalanx in the adult, as it ossifies well before the distal phalanges across taxa. Our study describes and revives the identity of the bony cap, and we identify it in three mammalian species: humans, cats, and horses (*Homo sapiens, Felis catus domestica*, and *Equus caballus*). During the embryonic period, we show the bony cap has a thimble-like shape that surrounds the proximal endochondral distal phalanx. The bony cap may thus play an inductive role in the differentiation of the corresponding nail, claw, or hoof (keratin structures) of the digit. When it is not present or develops erroneously, the corresponding keratin structures are affected, and regeneration is inhibited. By terming the bony cap, we hope to inspire more attention to its distinct identity and role in regeneration.

## INTRODUCTION

The identification of unique properties of the apex of the distal phalanx has been described since at least the Victorian era (*Dixey, 1881*; *Ewart, 1894*, *1895*; *Mettam, 1894*), but inconsistent naming or the lack of an alternative name for the apex muddles its significance. The "bony cap", as termed in this article, has been alluded to by other terms,

such as the ungule (*Hamrick, 2001*), ungual process or unguicular process (*Baran & Juhlin, 1986*; *Homberger et al., 2009*), or apical tuft (*Maiolino, Boyer & Rosenberger, 2011*), but these terms are not widely used and do not encapsulate the structure of the bone itself. In developing embryos, the bones of the limbs and the digits are endochondral and ultimately ossify during adulthood (*Hamrick, 2001*; *Stricker & Mundlos, 2011*; *Nakamura et al., 2016*). We posit an exception to this is the apex of the distal phalanges, the bony cap. The bony cap has a thimble-like shape that surrounds the distal phalanx and interacts with the developing distal phalanges and the nail epithelium. Complex tissue interactions such as the requirement of the nail or claw organ to be present for the regrowth of the distal phalanx bone may indicate that these distal zones are less understood than previously appreciated (*Zhao & Neufeld, 1995*; *Casanova & Sanz-Ezquerro, 2007*). Distal amputation and bone regrowth may require only the presence of the periosteum-derived (and therefore derived from mesenchyme) progenitor cells for regeneration (*Sensiate & Marques-Souza, 2019*), but transplant experiments show that input from osteogenic signals from the nail (ectodermal) may also be necessary and potentially sufficient for complete regeneration (*Mohammad, Day & Neufeld, 1999*). While it is known that most of the bone of the distal phalanx is mesenchymal and that the nail is ectodermal in origin, the origin of this intramembranous region between the two remains to be determined (*Casanova & Sanz-Ezquerro, 2007*). In this study, we explore more deeply across several mammalian taxa as to how the bony cap may be distinct from the distal phalanx and derived from the ectodermal wall or whether it develops in serial ossifications with the endochondral phalanges (*Homberger et al., 2009*; *Sensiate & Marques-Souza, 2019*).

The vertebrate skeleton is not entirely endochondral mesoderm. Certain bones are composites of endochondral and intramembranous components that eventually co-ossify (*Klíma, 1987*). In humans, the first indication of digits appears during the sixth week in the upper limb with the appearance of digital rays. The appearance of complete cartilage appears by the end of the sixth week (*Sadler, 2019*). The bony cap is clearly distinguishable from the distal phalanx early in development, but later it fuses with the distal phalanx as development progresses. The two often co-ossify into one. Consequently, we posit that the distal phalanx may have a dual nature: the intramembranous bony cap and the remaining endochondral distal phalanx. The bony cap may interact with two structures: the surface of the cartilaginous endochondral distal phalanges of the limbs, and the keratinous nails, hooves, or claws derived from ectoderm. The developmental origin of this structure is still unclear. If the bony cap ossifies prior to ossification of cartilaginous phalanges across multiple taxa, we suggest this provides further evidence of its ectodermal origin.

The intremembranous part is not unique to mammals and seems to occur across Amniota. There are clear implications for evolutionary modifications, although the mechanism of influence is unclear. Parts of the limbs are deeply conserved and yet plastic in adaptation. Units are not lost but interact in various ways (*Kavanagh et al., 2013*). We illustrate the presence of the bony cap throughout development and illuminate its morphology with three distantly related mammals with distinct forms: human (*Homo sapiens)* as the more plesiomorphic, domesticated cat (*Felis catus*) as the more specialized, and domesticated horse (*Equus caballus*) as the most complex due to their fused digits.

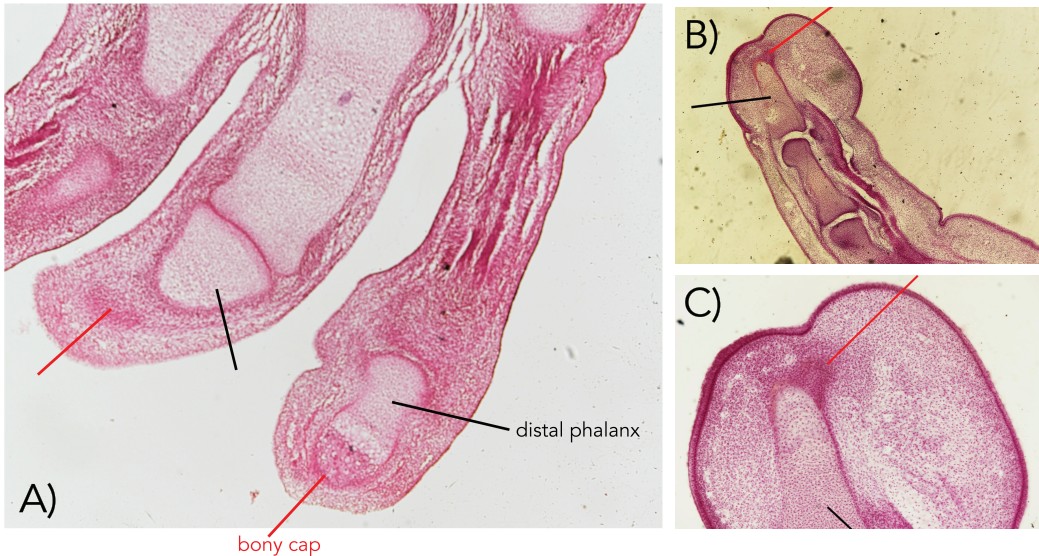

**Figure 1 Embryonic human fingers showing the bony cap (A–C).** The bony cap in a human from the Carnegie Collection of the Walter Reed Medical Museum (cap marked with red pointer; distal phalanx marked with black pointer). (A) Index and middle fingers from a 22 mm embryo in a longitudinal section (Carnegie Collection stage 22–embryo box number 62, 22 mm). (B and C) Index finger from a 50 mm embryo ventral is to the right, dorsal is to the left (11 weeks; Carnegie Collection embryo box 84 stage 22). Magnification of all 40×.                              

We demonstrate the bony cap as a distinct component of the distal phalanx and ultimately propose its role in mammalian bone regrowth. Regenerative biology of digits in higher vertebrates may also be applicable to medicine. Mouse studies have demonstrated that proximally amputated digits do not exhibit bone regeneration, while distally amputated digits do (*Sensiate & Marques-Souza, 2019*). When considering this, it may be possible that the bony cap is responsible for bone regrowth in amniotes.

Combining evidence from fetal and adult stages, we also synthesize our findings with those of two key studies: *Dixey (1881)* on the cat and human and *Ewart (1894, 1895)* on the horse. These studies are critical to our analysis because they are early works that describe this phenomenon, and we report them for the sake of reviving their findings. The dual nature of the distal phalanx is not unique, as other bones in the skeleton such as the skull, clavicle, scapula and the sternum contain both dermal and endochondral elements (*Klíma, 1968, 1987*; *Sánchez-Villagra & Maier, 2002*; *Fabbri et al., 2017*). Although the dual nature of the distal phalanx may be apparent and present across mammals and requires deeper investigation, this study brings awareness to this anatomical region and gives the intramembranous component a distinct name, as well as emphasis of its ubiquity despite extensive modification.

## MATERIALS AND METHODS

Using literature and visual inspection *via* multiple anatomical methods, three mammalian species were targeted for this study: humans *Homo sapiens* (Figs. 1, 2A, 2B, and 3), domesticated cat *Felis catus domestica* (Figs. 2C, 2D, 4, and 5), and domesticated horse *Equus callaus* (Figs. 2E, 2F, 6, and 7). Histology and micro-computed tomography (µCT)

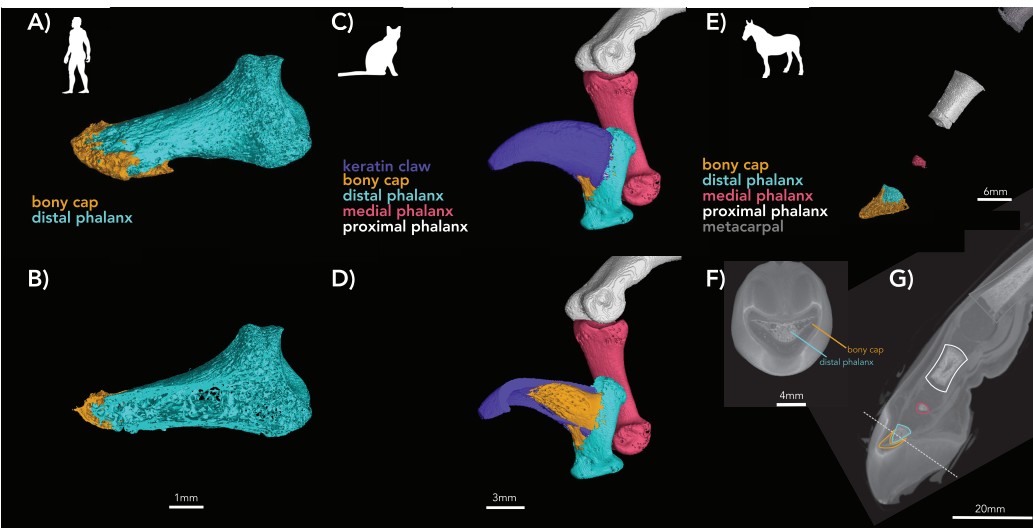

**Figure 2 μCT of digits of adult human, adult cat, and developing horse.** (A and B) Adult human pollex (digit I) (A) external surface with bony cap and distal phalanx segmented; (B) internal view of the same specimen. The bony cap is a thin veneer. (C and D) Adult cat (digit IV); (C) external surfaces of phalanges and keratin claw segmented. The bony cap articulates with the distal phalanx folds, and this process of the distal phalanx is wedged into the claw. The keratinous sheath folds over the dorsum and covers the left and right sides of the bony cap. (D) Internal view of the same specimen. (E and F) DiceCT of an embryonic horse manus digit with phalanges and bony cap segmented. The images are shown twice to facilitate color-coding and present the scans intact. (E) Lateral external surface of the manus showing the bones of the metacarpal, proximal, middle, and distal phalanges partially ossified and (F) a distal transverse section. (G) Shadow image of the entire structure. The ossification of the distal phalanx, middle phalanx, yellow pointer, and proximal phalanx has taken place.

were performed on both adult and embryonic specimens and details are outlined below. In inspection of the histological slides, the bony cap was designated as separate if we could identify evidence of condensing cells moving from a distinct direction from the phalanx and clear physical separation. In inspection of the μCT-scans, the bony cap was identified through segmenting of distal bony regions that were completely distinct without direct articulation with the more proximal bony tissue or cartilage. Once reconstructed, we could make clear how these individual slices surrounded the underlying tissue. We emphasize our reconstructions are hypothesized boundaries, and that formal molecular markers of cellular boundaries and barriers are necessary for more accurate evidence of distinct tissues.

## Human

The bony cap in human embryos (Fig. 1) was investigated from specimens of the Carnegie Collection of the Walter Reed Army Institute of Research—National Museum of Health and Medicine. The specimens were studied under normal light at 20× and 40×. Examination of embryos smaller than 22 mm was inconclusive for the presence of the bony cap, and thus only specimens older than this were included. We selected one index and one middle finger from a 22 mm embryo (stage 22; Carnegie Collection stage 22—embryo box number 62, 22 mm) and the index finger from a 50 mm embryo

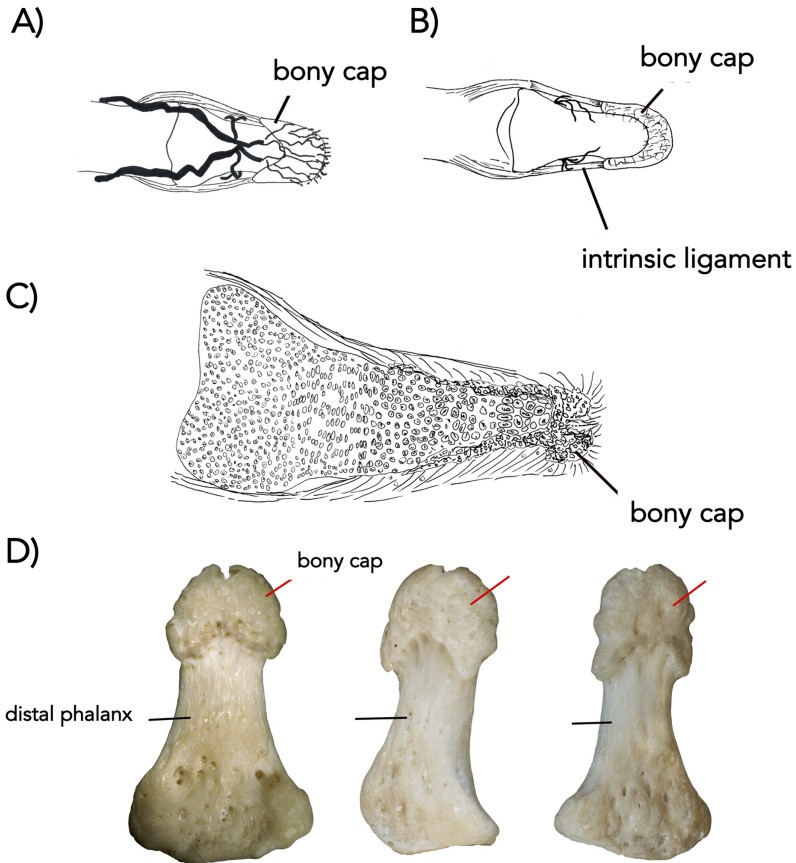

**Figure 3 The bony cap fused onto the distal phalanx of adult humans (A–D).** (A) Arteries, distal phalanx and bony cap redrawn from *Flint (1955)*. (B) Early human embryonic bony cap and distal phalanx re-drawn from *Dixey (1881)*. (C) Three human thumbs from NYIT College of Osteopathic Medicine collection (cap marked with red pointer; distal phalanx marked with black pointer).

(11 weeks; Carnegie Collection embryo box 84 stage 22) and performed traditional histological preparation with hematoxylin and eosin stain. We also μ-CT scanned three human adult distal phalanx specimens from the collection at New York Institute of Technology College of Osteopathic Medicine (Figs. 2A and 2B) using the Yale University high-resolution Nikon H225 ST μCT scanner at a voxel size of 0. 0.03223787 mm, 149 kV, and 98 μA. Scans were reconstructed using in-house Nikon software protocols and analyzed and segmented in all 3D planes using VG Studio Max v. 3.4.

## Cat

A fetal cat (body length 30 mm; NS 294) was purchased from Ward's Biological Supply and imaged using the NYITCOM μ-CT under the following parameters: 45 kV, 177 μA, 2K resolution, 0.002 mm Copper exposure filter, 6.4 spot size, 0.3 μ-CT ns, rotation steps, seven frames, 20 random movements, 360-degree scan. The left pes of an adult domesticated cat (NS 292) was μCT-scanned at Yale University under parameters of

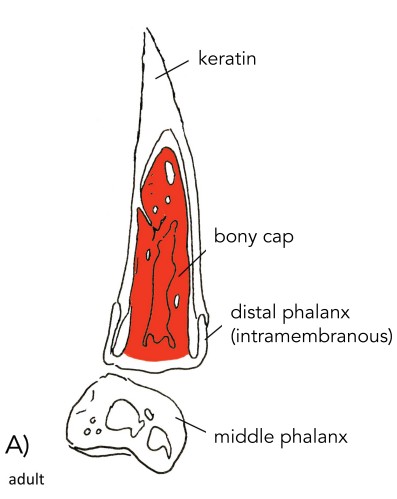
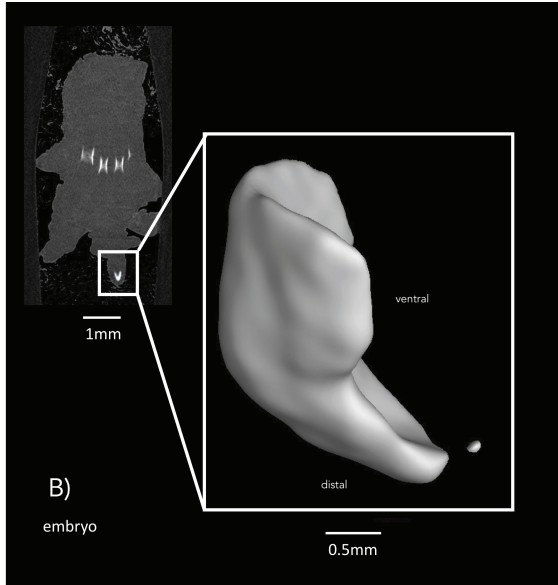

**Figure 4 The ungual process of bony cap in a foot of a domesticated cat (A and B).** (A) Specimen (NS 292) was scanned by μCT (A) adult, cap marked with red; distal phalanx and middle phalanx with no color (left foot digit IV). (B) Segmented bony cap from μCT-scan a young fetal cat (finger IV) (body length 30 mm—NS 293) from Ward's Biological Supply. At that stage, the phalanges were cartilaginous and not observable. The other visible ossified regions are the developing metacarpals. The view of the scan is a transverse view of the scan and the region that was segmented and reconstructed is shown in the smaller rectangle. The larger rectangle shows the reconstructed 3D mesh of the bony cap, rotated to show the medial side for clarity. We have labeled the views of the 3D region for clarity.

0.05919933 mm voxelsize, 150 kV, and 98 μA. Processing was similar to that of the human. The fourth digit of the pes was selected for figures (Figs. 2C, 2D, and 5).

## Horse

We used three samples of three horse fetuses purchased from Ward's Biological Supply. The whole-body length in millimeters was measured with digital calipers from frontal bone to rump. A larger near-full term horse was sectioned with a saw. The pes (NS 293; 762 mm whole body length) was removed and μCT-scanned at Yale University using 0.06574807 mm voxelsize, 81 kV, and 120 μA. The specimen was then stained with 10% concentration Lugol's iodine (I₂KI), which allowed for visualization of the soft tissue (*Gignac & Kley, 2014*). The stained limb was imaged at Yale University using the 0.0636894 mm voxelsize, 81 kV, and 66 μA. Yale scan data was processed and visualized as described for the human. A developing horse (NS 292; 210 mm whole body length) was μCT-scanned at NYIT using similar parameters above, and another near-full term horse (NS 801; 750 mm whole body length) was sectioned longitudinally. Figures 6A and 6B were redrawn after published figures and histological specimens described by *Ewart (1894, 1895)*. The other manus from the specimen of Fig. 2, was sectioned transversely near the proximal, middle, and distal end (Fig. 7).

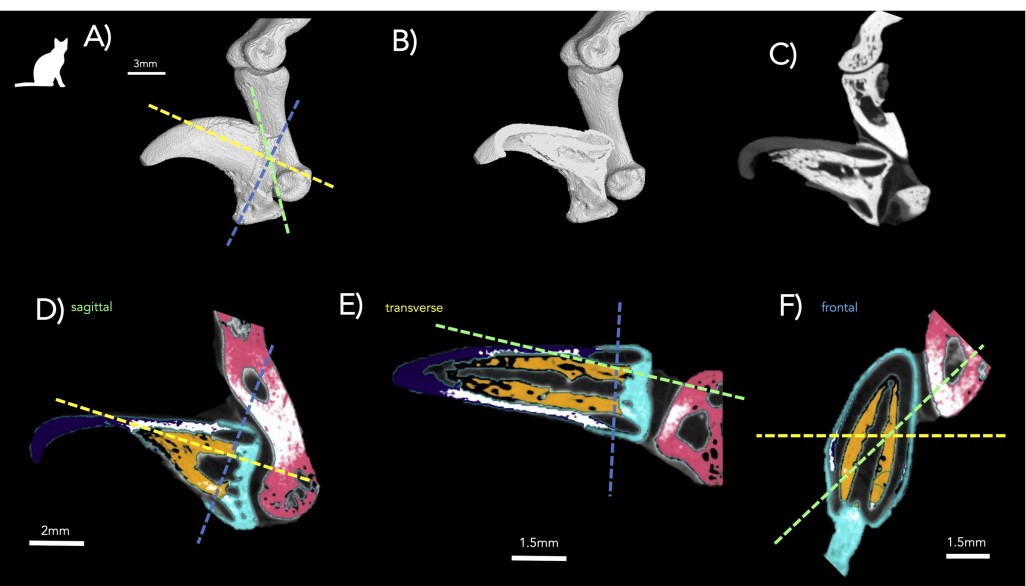

**Figure 5 Detailed view µCT-scans of cat pes to demonstrate segmentation boundaries.** (A) 3D external view of digit IV with colors corresponding to anatomical planes shown in D–F. (B) 3D internal view of 5A; (C) 2D sagittal view of 5B; (D) sagittal segmented structures showing middle phalanx (pink); distal phalanx (teal); bony cap (yellow); and keratinous claw (purple); (E) transverse view of segmenting; and (F) frontal view of claw, where distinct boundaries of the bony cap from the phalanx are visible.

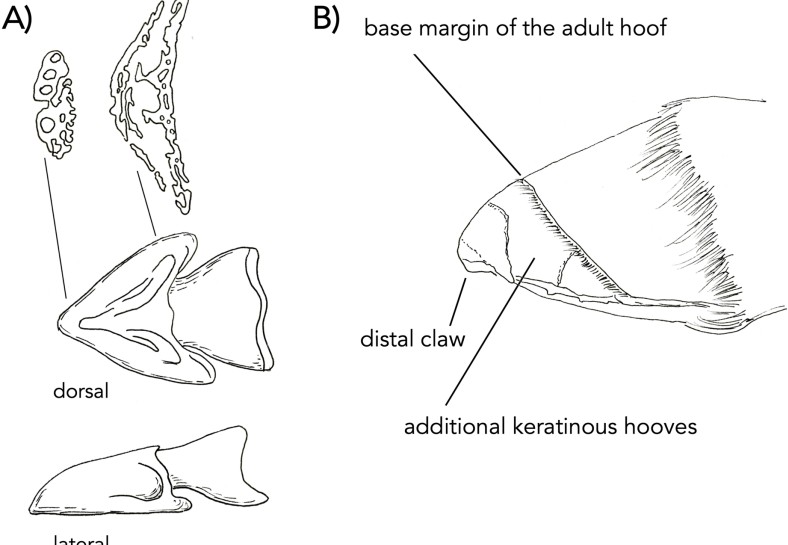

**Figure 6 Early development of the distal phalanx of a domesticated horse.** (A) Dorsal and lateral view of embryonic bony cap redrawn after published figures and histological specimens described by *Ewart (1894, 1895).* The transverse sections are drawings from our own data (µCT) from Fig. 2. The distal section shows five bony cap tubes for the five digits. The proximal section shows three spaces for the more proximal digits. The proximal section shows three chambers: a median one for digit III and the other two presumably related to other digits (I and V). Eventually, the entire structure will fill with endochondral bone and co-ossify. (B) A figure re-dawn from *Ashdown & Done (1987)*; figure 7.81 of a newborn foal foot. The horse is born with a claw structure and multiple keratinous surfaces which eventually wear off while the main hoof remains.

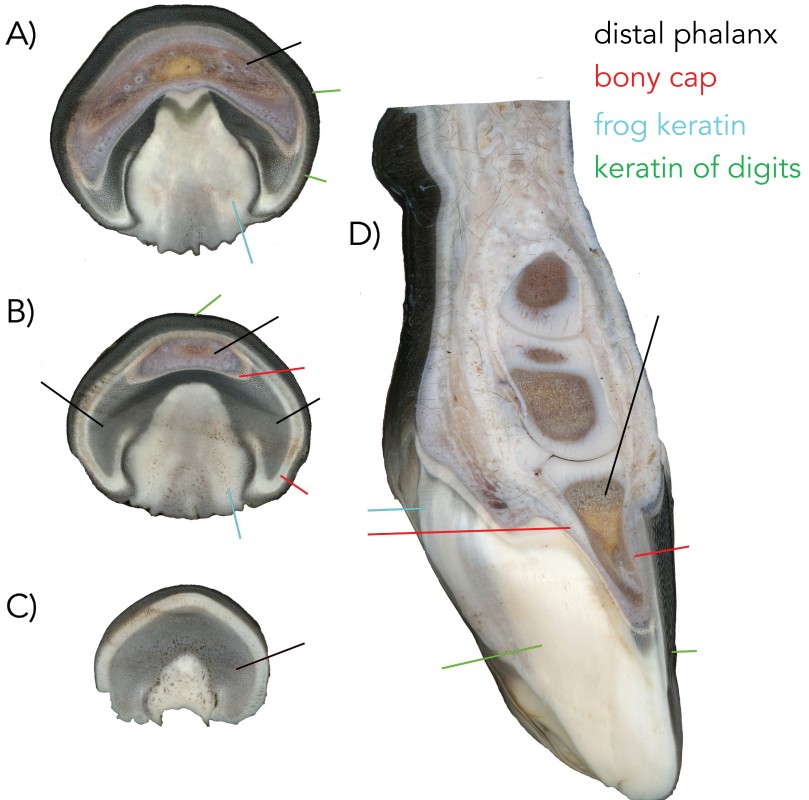

distABDistal phalanx
bony cap
frog keratin
keratin of digits

**Figure 7 The manus of a fetal horse near full term (A–D).** (A) Proximal transverse section showing the frog which are interpreted as parts of digits II and IV. Green pointer shows the wings which are interpreted as digits I and V. Black pointer shows the bone of digit III. (B) Section through the middle of the phalanx. Black pointer shows the bone of digit III, I, and V. Green pointer shows the keratin of digit III. Blue pointer shows the frog which is keratin of II and IV. The red pointer shows the still distinct bone of the bony cap. (C) Distal transverse section showing the bone of digit III (black). (D) Longitudinal section through the manus of the same individual (manus of other side). Black is the distal phalanx of digit III. Red pointers show the bony cap which is intimately congruent to the distal phalanx. Green pointer is the keratin of III. Blue pointer is the keratin of the frog.

## Data availability

The specimens are available in the NYIT repository. All processed μCT-scans are deposited to Morphosource (https://www.morphosource.org/projects/000387927). The following scans of the phalanx or distal pes are available under the following ARK identifiers: adult human pollex (NS_human_specimen4: http://n2t.net/ark:/87602/m4/391069), cat embryo (NS294: http://n2t.net/ark:/87602/m4/391064); adult cat (NS292: http://n2t.net/ark:/87602/m4/389105), horse embryo (NS293: http://n2t.net/ark:/87602/m4/389086), iodine-stained horse embryo (NS293: http://n2t.net/ark:/87602/m4/389095). Our team has elected to re-draw some of the figures from μ-CT, and these segmented images are available on Morphosource under the following identifiers: NS293 embryo horse ungule (http://n2t.net/ark:/87602/m4/417487), NS294 embryo cat ungule (http://n2t.net/ark:/87602/m4/419934), NS292 adult cat ungule (http://n2t.net/ark:/87602/m4/419937), and NS_human_specimen4 ungule (http://n2t.net/ark:/87602/m4/419940).

## RESULTS

The bony cap was located in the fetal (after 22 mm) and adult specimens of humans, domesticated cats, and domesticated horses. It had a similar crescent appearance in all three species, was positioned at the tip of every digit, and had clear ossification patterns prior to that of the distal phalanx. A description is given for each morphology and species.

### The bony cap in the human fetus

The bony cap can be distinguished from the endochondral distal phalanx as early as 14 weeks of development. We find it important first to synthesize information that has been described over a century ago to help put our results into context. From the literature (*Dixey, 1881*), the description of the bony cap in the developing human is as follows: "Near the tip of the distal phalanx, there is a region where the cartilage changes. The cell spaces are smaller, and the matrix is more deeply stained than the sides surrounding it, resembling the characteristics of primary bone with osteoblasts and osteogenic fibers. This distinguishes the presence of primary bone when at this point in development the endochondral distal phalanx is cartilage. The bony cap has a pit between its dorsal and ventral surfaces, where the dorsal surface has far fewer lacunae than the ventral side". The bony cap can be observed in longitudinal sections of the digit (*e.g.*, *Krstić (1991)*, plate 220). We corroborate these descriptions with our data, such that the index and middle fingers show representative typical bony caps in the distal region of the distal phalanges. This is best observed in a 22 mm embryo (stage 22) (Fig. 1A). In older fetal specimens (50 mm embryo (11 weeks); Figs. 1B and 1C), the distal phalanx resembles more that of an adult as it becomes more ossified. The apical cap is enlarged on the distal ventral side as it is in the adult.

### The bony cap in the adult human

In the adult human skeleton, the bony cap is demarcated by the rough area near the apex of the distal phalanx. Unlike the embryo, the surface of the distal phalanx is smooth. It is horseshoe shaped and well developed ventrally. Dorsally, it surrounds the periphery of the apex. Given the presence of the same structures in all four digits we scanned, we report it is likely found in all 20 human digits (Fig. 2A) and coats the phalanx like a thin veneer (Fig. 2B). Intraosseous ligaments extend from tubercles on the lateral aspects of the base of the distal phalanx to spines at proximo-lateral apices of the tuft-like portion at the tip of the distal phalanx (*Shrewsbury & Johnson, 1975*), and they create two tunnels between the ligaments themselves and the distal phalanx. The arteries pass through these tunnels, contributing to the large vascular bundle that supplies the nail bed and passes ventrally (Figs. 3A and 3B).

### The bony cap in the fetal domestic cat

*Dixey (1881)* also studied a 40 mm cat embryo, which showed unaltered cartilage at the proximal base of the phalanx and an incomplete thin sub-periosteal cap of intramembranous bone. There are osteoblasts and fibers at its outer borders and lacunae inside the core of the section. A strong remodeling zone of the bony cap is present. In the

domesticated cat embryo, the bony cap has an elongated crescent shape with a distinct pointed distal end and a hollow center (Fig. 4), which may be the keratinous claw forming dorsally to the bony cap and adopting its hooked and pointed shape. The µ-CT imaging of the 30 mm cat embryo displays ossified bony caps on all developing digits. However, at this embryonic stage and µ-CT resolution, the cartilage model for the developing endochondral phalanx was not visible (Fig. 4B).

### The bony cap in the adult domestic cat

The adult bony cap and distal phalanx are very similar to those of the late fetal specimens. The difference is that both structures are ossified and fused to each other (Figs. 2C, 2D, 4A, and 5). The bony cap is situated ventral to the keratinous claw (Figs. 2D and 5B–5E). Unlike in the human and horse, the bony cap of the cat is hollow (Figs. 5E and 5F). This is because the distal phalanx does not insert into the bony cap as it does in other species. We deduced that the distal phalanx is a specialized bone with a deep median groove (Figs. 2C and 2D). Inside that groove, the thick hollow proximal end of the bony cap fits congruently and is fused in the adult (Figs. 4A, 5D, and 5E). The atypical shape may reflect the specialization for retractable claws, as these large sesamoids allow for the flexor tendons.

### The bony cap in the fetal horse

*Ewart (1894, 1895)* are thorough discussions of the forelimb of the horse, including detailed figures drawn from high-resolution histology. The focus is on the third digit, which is well known to be dominant in horses. He begins by describing embryo A (20 mm in length) and ends with a discussion of embryo H (965.2 mm in length). Reviewing from the literature, we will report embryos A–C from these studies, with an emphasis on embryo C. In embryo A (20 mm in length; approximately 30 days old), the forelimb at this stage is entirely cartilaginous. The distal phalanx has formed. The trapezium is absent at this point, but Ewart suggests this may be a product of the dissection. The magnum is also noted to be narrower than in the adult. At this point, Ewart does not refer to the bony cap. In embryo B (25 mm in length; 40 days old), the forelimb is now longer but remains completely cartilaginous. The splint bones are shorter than the third metacarpal, closely resembling the adult. Ewart does not refer to the bony cap at this stage either. In a section near the distal ends of metacarpals II and IV, II appears to be larger than IV. In embryo C (50 mm in length; unspecified age), at this stage, the bony cap is present and fits over nearly half of the cartilaginous distal phalanx. We posit here that the bony cap is intramembranous in origin, while the distal phalanx is endochondral. Ewart extensively described the complexity of the bony cap and reported a central conical part, an irregular terminal end, and two wing-like expansions (Fig. 6).

The µ-CT imaging of specimen NS 294 is consistent with the findings that Ewart described. *Solounias et al. (2018)* also showed that the forefoot of the domesticated horse is composed of five digits. Our data shows the distal-most part of the bony cap as having five

tubes lined up next to each other. More proximally, there are only three tubes (Fig. 7), with a central tube and two lateral irregularly shaped tubes (Fig. 7). Thus, every digit has a bony cap and the main digit III has the largest and most complete bony cap. Moving proximally, the bony cap terminates. Proceeding proximally along the pes (Figs. 2E and 2F), there are punctuated areas of bone where ossification of the distal phalanx, middle phalanx, and proximal phalanx has taken place. The features distinguish the bony cap (yellow) at this stage is recognizable as a separate structure from the distal phalanx proper, though still surrounding the distal phalanx like a "cap" (Fig. 2E, 2F, and 7).

When a foal is born, it always has a claw at the apex of the hoof. It also has additional surfaces of keratin which reflect the more-than-one digits merged. These structures wear off as the animal runs resulting in an adult hoof. Images of the additional keratins and claw are visible in a drawing (Fig. 7C after—*Ashdown & Done (1987)*; fig. 7.81). The mid-sagittal section of the front left hoof of a near full term fetal horse (762 mm; Fig. 7) (NS 293) displays a white line that outlines the distal phalanx where the keratin comprising the hoof originates. It starts at approximately the mid-point of the distal phalanx and surrounds it dorsally and ventrally. We argue that this white outline is the bony cap (Fig. 7). The three transverse sections show the frog (Figs. 7A–7C), which is interpreted as parts of digits II and IV. The green pointers show the wings, which are interpreted as digits I and V. The black pointer shows the bone of digit III. A sagittal section through the phalanx (Fig. 7D) is also showing the various keratins. The black pointer shows the bone of digit III, I, and V. The green pointer shows the keratin of digit III. The blue pointer shows the frog which is keratin of II and IV. The red pointer shows the still distinct bone of the bony cap, which is intimately congruent to the distal phalanx. The black pointer is the distal phalanx of digit III. This is critical, as it will co-ossify with the distal phalanx that has penetrated it.

## The bony cap in the adult horse

The distal phalanx of the horse has two types of surfaces: a rough one that binds to the keratinous hoof and a smooth one that is proximal to the rough and is free of keratin covers. The corium is situated deep to the keratin. We postulate that the rough portion is a thin veneer of bone that is a remnant of the adult bony cap, and it surrounds the proximal distal phalanx. The smooth surface and most of the interior of the remaining distal phalanx is embryologically endochondral. The two co-ossify. Most of the distal phalanx is comprised of digit III, flanked by digits I and V (*Solounias et al., 2018*). The clear separation between the distal phalanx and bony cap is not present in adult horse, as there is fusion and remodeling of the bones; however, the two can be distinguished by the border between the smooth and rough surfaces. The shape of the bony cap is congruent to the overlying keratin, and it has both dorsal and ventral areas with the ventral side being most pronounced. Its pitted rough surface is less pronounced on the ventral side than dorsally, potentially due to the pressure the horse places on the ventral aspect of its hoof. There is a pronounced groove demarcating the distal phalanx from the bony cap on the ventral aspect of the digit. Four foramina are visible on the proximal side between the two bones for the digital arteries, two medially and two laterally. The arteries penetrate the distal phalanx, as it represents the fusion of three digits into one.

## DISCUSSION

Since the late 1800s, scientists have noticed distinguishing features about the distal-most part of the mammalian digits (*Dixey, 1881*; *Ewart, 1894*, *1895*). More recently, literature has shown the bony cap but has not identified it as different from the remaining distal phalanx (*Baran & Juhlin, 1986*; *Homberger et al., 2009*). We describe the bony cap in humans, cats, and horses. In humans and horses, the bony cap has a crescent shape that couples and surrounds the distal phalanx, reminiscent of a horseshoe, that is mostly ventral and lateral. In these two species, the bony cap is enveloping the distal phalanx and will fuse with the phalanx in adulthood. In cats, the shape is crescent-like due to the specialization of retractable claws. Because of its' specialized features, the orientation of the bony cap to the distal phalanx in the cat is modified and thus the bony cap is hollow centrally. The bony cap is observed in all digits we analyzed, and we predict it is present in all digits, pes and manus alike.

### The bony cap has an embryological origin distinct from the cartilaginous distal phalanx

The significance of our work is the recognition of the distinct embryological history of the bony cap in relation to the cartilaginous remaining distal phalanx. *Grüneberg & Lee (1973)* discovered that the autosomal recessive gene for brachypodism results in basal and middle phalanges that are abnormally ossified while the terminal phalanges are normal or nearly normal. They conclude that this is because "much of the terminal phalanx is formed directly from membrane and not by replacement of its cartilaginous 'model' on which it sits like a thimble." These findings further hint at the unique embryological history of the bony cap. The induction of the intramembranous bony cap by signaling molecules *Shh*, *Hoxa13*, *Bmp2*, *Bmp4*, *Wnt7a*, *Msx1*, and *Msx2* is discussed as the chondrogenic condensations at the terminal phalanges (Table 1 of *Hamrick, 2001*). Although especially present in the cat, the bony cap is present and ossified in the embryo prior to the ossification of the distal phalanx across species.

The complex nature of the distal phalanx and the bony cap is frequently misunderstood or incompletely described. Common knowledge of endochondral ossification includes the development of a primary ossification center, with one or more secondary ossification centers developing later (*Williams et al., 1995*, fig. 650). Early descriptions speculate that the conical shape of the endochondral distal phalanx in horses and humans is due to its growth being restricted by the ossified bony cap (*Dixey, 1881*; *Ewart, 1894*). A previous study explains that the distal end of the distal phalanx has an intramembranous origin, but we propose the intramembranous bone cannot be considered an ossification center for endochondral bone differentiation (*Kamibayashi et al., 1998*). Rather, the three ossification centers described may be the distinct intramembranous ossification of the bony cap and the separate endochondral ossification of the proximal distal phalanx. A previous study in a 10-day-old kitten found the presence of a growth plate between the articular cartilaginous base and the bony cap, which they termed the "unguicular process" (*Homberger et al., 2009*). There is a distinct separation of the two bones embryologically,

particularly given that the bony cap appears to ossify prior to the distal phalanges, suggesting it may have a distinct ectodermal origin rather than the mesenchymal phalanx bone. There are other curious features of the cat claw that are also important to note here. *Homberger et al. (2009)* depicts that the growth plate of the distal phalanx is located distal to the flexor tubercle which indicates that the tubercle is part of the phalanx that ossifies endochondrally. *Homberger et al. (2009)* also indicates that the unguicular hood (which extends from the endochondrally derived portion of the distal phalanx) is ossified intramembranous several months after birth, but the authors do not indicate anything similar for the flexor tubercle. Given that we found ossification of the bony cap early in development, the bony cap in the embryonic cat is distinct and not derived from the phalanx (Figs. 1, 4, and 6). We posit the flexor tubercle and unguicular hood are both derived from the distal phalanx, though given the delayed ossification of the unguicular hood (outside the scope of this study), it is possible this is a third distinct structure that warrants further investigation or there could be variation across cat breeds. Corroborating *Dixey (1881)*, we refute an endochondral template shaping the bony cap and show that the bony cap is already ossified before the endochondral distal phalanx. Thus, the unique shape of the distal phalanx relative to the other phalanges may be due to the presence of the bony cap. This may cause the proximal endochondral distal phalanx to grow laterally. The shape of the distal phalanx may also be modified in cats by the presence of the bony cap, but because of the specialization of the claw, it shows a modified proximal articular base (*Homberger et al., 2009*). Thus, we posit that the distal phalanx has a dual nature of embryological origin. This is a speculation that needs experimental verification. However, adjacent forming organs are known to interact in a complex back-and forth-signaling and with potential constraint to the adjacent tissue (*e.g.*, between tooth and jawbone, *Renvoisé et al., 2017*).

## The bony cap shows species-specific variation

Although present in the three species, the bony cap does have unique features in each. Humans indeed possess the simplest type of a bony cap. There are two interosseous ligaments intrinsic to the distal phalanx (*Flint, 1955*). Ligaments are connective tissue commonly found between two separate bony structures; however, in atypical bones like the scapula and clavicle, ligaments can be found connecting different parts of the same bone. We suggest that the presence of these intrinsic ligaments in the distal phalanx is further evidence of the unique embryological development of the bony cap. In the cat, the bony cap is rotated and co-ossifies in a unique way to the distal phalanx, which may be a specialization for hooking its prey. Its bony cap is elongated and is very similar to the shape of the claw. The 38-day-old cat fetus showed that at this point in gestation, the bony cap was the only ossified structure in the developing limb (Fig. 4B). The same pattern of ossification of the distal-most phalanges occurs in mouse limb development (*Fleckman et al., 2013*), suggesting the developing endochondral distal phalanx will ossify in a manner to accommodate the already present bony cap. The cat bony cap also remains hollow due

to the distinct orientation of the distal phalanx relative to the bony cap. In domesticated horses, the bony caps are multiple because the horse hoof consists of the integration of five digits into one (*Solounias et al., 2018*; *Kavanagh, Bailey & Sears, 2020*). In both the horse and human, the bony cap becomes filled with the distal phalanx and both will fuse into one entity. The bony cap in the cat will fuse in adulthood to the distal phalanx, but never covers it as it does in the other species.

## The role of the bony cap in nail formation

We postulate that the bony cap may play a significant role in nail formation in humans, claw formation in cats, and hoof formation in horses. While this hypothesis has been proposed in platyrrhine primates (described as the apical turf (*Maiolino, Boyer & Rosenberger, 2011*)), its ubiquity across mammals (and even tetrapods) needs further consideration. Using inference from the rare congenital ectopic nail in humans, appropriate nail formation is due to the interaction between the distal phalanx and the nail, and it has been demonstrated that bone deformities are associated with alterations of the nail (*Baran & Juhlin, 1986*; *Kamibayashi et al., 1998*). Because the bony cap specifically interacts with the nail, it may allow for its differentiation. In the cat, the keratinous claw is open ended, resembling a taco shell (Figs. 2B, 2C, and 5). The claw extends directly distal from the bony cap and is shaped congruently to the bony cap. In a figure detailing the hoof of an unborn foal, five divisions can be seen in the softer more embryonic keratin (Fig. 7), reflecting partial separation of the five fused digits in the horse hoof (*Solounias et al., 2018*). These five keratin structures are likely due to the five bony caps, as present in our μCT-scans (Fig. 6 is drawn representation), found at the distal parts of the five digits within the hoof. In this figure, the harder more proximal keratin is the embryonic precursor to the adult hoof. The separation between the distinct bony caps is harder to decipher as compared to the adult hoof because of the partial differentiation.

## The role of the bony cap in regenerative biology

There are numerous studies that examine the regeneration of the bone in the distal phalanges. *Zhao & Neufeld (1995)* reported that the removal of the nail organ prohibited bone regrowth and concluded that the nail organ profoundly impacts bone regrowth. There may be a distal influence in bone regeneration located near the nail, which may indeed be the bony cap without realizing this. Further, bone regeneration at the distalmost aspect of the distal phalanx has an inductive role in bone regrowth (*Mohammad, Day & Neufeld, 1999*; *Sensiate & Marques-Souza, 2019*). The location of the bony cap correlates with the findings in these studies and this region could be the region responsible for bone growth. It is possible that this intramembranous bony cap region could house stem cells that would likely be involved in the regeneration process. The recognition of the bony cap with a unique name and describing it in three species reinvigorates its identity and illuminates a promising link with regenerative biology. The fascia of the distal phalanx is complex and likely interacts with multiple surrounding structures (*Shrewsbury & Johnson, 1975*).

## CONCLUSIONS

We describe distinct developmental ossification of a structure termed the "bony cap" on the distal phalanx of three mammalian species. The variation in the structure among the human, cat, and horse suggests the structure is subject to selection that facilitates extensive modification for divergent limb functions. Future investigations on both its ubiquity in tetrapods, as well as its role in regenerative bony tissue, will clarify a deeper understanding of the form and function of a morphological structure that has been largely ignored for most of the last century.

## ACKNOWLEDGEMENTS

We thank Elizabeth Locket at the Walter Reed Army Institute of Research – National Museum of Health and Medicine, where we had access to top-of-the-line microscopes and the famous Carnegie Collection of embryos. We thank the departments of Mammalogy and Paleontology at the American Museum of Natural History for specimens. We thank our librarian Mahnaz Tehrani and Matthew Mihlbachler at the NYITCOM. We also thank Kelsi for the μ-CT imaging. We also thank Brandon Mercado and Anjan Bhullar at Yale University for assistance in μ-CT scanning that took place at Yale.

### Funding

This work was supported by the National Science Foundation (NSF) MRI-1828305, NSF-PRFB 1812035 Postdoctoral Fellowship in Biology, and the NYITCOM Academic Medicine Scholars Program. The funders had no role in study design, data collection and analysis, decision to publish, or preparation of the manuscript.

### Grant Disclosures

The following grant information was disclosed by the authors:
National Science Foundation: MRI-1828305, NSF-PRFB 1812035.
NYITCOM Academic Medicine Scholars Program.

### Competing Interests

The authors declare that they have no competing interests.

### Author Contributions

- Shannon Smith conceived and designed the experiments, performed the experiments, analyzed the data, prepared figures and/or tables, authored or reviewed drafts of the article, and approved the final draft.
- Laurel R. Yohe performed the experiments, analyzed the data, prepared figures and/or tables, authored or reviewed drafts of the article, and approved the final draft.
- Nikos Solounias conceived and designed the experiments, analyzed the data, authored or reviewed drafts of the article, and approved the final draft.

## Data Availability

The specimens are available in the collections of the New York Institute of Technology College of Osteopathic Medicine.

All processed μCT-scans are available at Morphosource: 000387927. Our team has elected to re-draw some of the figures from μ-CT, and these segmented images are also available within this project.

https://www.morphosource.org/projects/000387927.

The scans of the phalanx or distal pes are available at Morphosource:
- adult human pollex, NS_human_specimen4, 10.17602/M2/M391069;
- cat embryo, NS294, 10.17602/M2/M391064;
- adult cat, NS292, 10.17602/M2/M389105;
- horse embryo, NS293, 10.17602/M2/M389086;
- iodine-stained horse embryo, NS293, 10.17602/M2/M389095.

The segmented images are available at Morphosource:
- embryo horse ungule, NS293, 10.17602/M2/M417487;
- embryo cat ungule, NS294, 10.17602/M2/M419934;
- adult cat ungule, NS292, 10.17602/M2/M419937;
- *Homo sapiens*, NS_human_specimen4 ungule, 10.17602/M2/M419940.

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
