# Peer review of "The bony cap and its distinction from the distal phalanx in humans, cats, and horses"

_PeerJ, doi:10.7717/peerj.14352_

## Round 0.1 · original submission · Major Revisions

The reviewers raise a number of excellent and constructive points that seem eminently worth addressing. In addition, I'll add the following for the Data Availability section.

1. It appears the specimens were procured from a vendor, and are not being preserved in a museum or other collection but rather, presumably, in the lab of one of the authors ("available upon request"). Experience shows that specimens kept outside of a collection dedicated to preservation are at significant risk of getting lost over time. In the interest of facilitating reproducibility, please consider making the specimens available through a suitable collection under a collection accession number (identifier), or explain why this is not possible.

2. It appears Morphosource does not currently support searching projects by identifier. Therefore, please reference the full URL for the project (https://www.morphosource.org/projects/000387927) rather than just the identifier.

3. In the interest of reproducibility, please deposit the redrawn segmented images (referenced currently as "can be requested") in a suitable data repository (such as Dryad) and give the corresponding record identifier(s). If for some reason this can't be done, please explain.

4. The ARK identifiers appear hyperlinked but show as ark:/<ID>. Browsers and other common identifier resolution tool are unable to resolve identifiers in this form. Please consider giving them instead with their resolvable n2t.net HTTP prefix (or that of another ARK resolver), which should yield IDs resolvable by copy&pasting to a web browser.

Reviewer 1 ·

Basic reporting

1) I believe a description and synthesis of the process of distal phalanx development during embryogenesis and subsequent growth in the introduction would greatly facilitate the accessibility of this paper as well as to clarify the bases of some of their interpretations. Specifically, it would be helpful to specify what is known in regards to the relative timing of appearance of tissues and ossification. Further, the authors may want to discuss how the bony cap may or may not differ from the periosteal cuff that forms during endochondral ossification of long bones, but differs from the primary ossification center because it grows appositionally rather than endochondrally.
1.a. The authors briefly mention in lines 51-53 that “The embryology of the bony cap is different from the remaining distal phalanx, as it is derived from the ectodermal wall (Zhao & Neufeld, 1995; Homberger et al., 2009; Sensiate & Marques-Souza, 2019),” but I feel this requires further clarification. While it is not contested that the different portions grow in different manners, the references they have cited do not indicate an ectodermal origin for the intramembranous portion. Can the authors clarify? Is there a missing citation? Following Zhao and Neufeld’s study on regrowth after digit amputation, nail induced bone growth does not necessarily mean that the bone is embryologically derived from ectoderm, rather it would seem that the regrowth of the bone is influenced by interaction between two tissues of different embryological origin: mesenchyme (bone) and ectoderm (nail/claw). Sensiate & Marques-Souza (2019) provide further evidence that during regrowth, signals that activate appositional ossification emanate from nail epithelium, but that the actual osteoprogenitor cells are located within the periosteum of the remaining portion of the distal phalanx.
1.b. Similarly, the authors mention a “dermal origin” for the bony cap of the cat in lines 286-287, but it is not clear what this means. The dermis anchors the claw to the distal phalanx and forms the claw bed, but I don’t think the intramembranous bone of the distal phalanx originates from it embryologically.
2) It would be helpful if the authors were to clarify their anatomical descriptions by referring to more specific structures of the distal phalanges such as the articular surface, extensor tubercle (insertion for long digital extensor tendon/apparatus), and flexor tubercle (insertion for long digital flexor tendon).
2.a. This most strongly applies to the description of the cat distal phalanx and its parts. The authors describe the endochondrally derived proximal portion as having a curved L-shape with a deep median groove and refer to Figure 4 in explanation. In Figure 4, the bony cap of an adult cat is highlighted in red and this is shown to include the articular surface while part b of the same figure has a line pointing to the flexor tubercle with the text “distal phalanx (intramembranous).” It is not clear to me what the L-shaped portion with a median groove actually refers to. Is the bottom of the “L” the flexor tubercle and the long limb the base/articular facet? If this is the case, why is the bony cap (as indicated in red) incorporating the articular facet while the flexor tubercle is pointed to and labeled as intramembranous? This is further confused by the indication of the bony cap in Figure 5, in which the red portion highlighting the cap contrasts that of Figure 4 by excluding the articular facet of the distal phalanx. It would be helpful if the authors clarified their descriptions and ensured that their depictions across figures are consistent. It should also be noted that Figure 4 of Homberger et al 2009 depicts that the growth plate of the distal phalanx is located distal to the flexor tubercle which indicates that the tubercle is part of the phalanx that ossifies endochondrally. Homberger et al 2009 does indicate that the unguicular hood (which extends from the endochondrally derived portion of the distal phalanx) is ossified intramembranously several months after birth, but they do not indicate anything similar for the flexor tubercle.
3) A major goal of this paper is to document the anatomy of the distal portion of the distal phalanx that resulted from intramembranous ossification during growth and development. For figures in which the bony cap and/or rest of the distal phalanx has been segmented from a microCT scan, this goal would be better achieved by showing the segmented distal phalanx in different views as opposed to simply one.
3.a. This is especially relevant to Figure 5, part b in which the embryological bony cap of a cat is depicted in a single view. What view is this in? That information should be included in the figure caption. Providing additional views of this element would permit the appreciation of its three-dimensional structure.
4) Figures 7 (line 118) and 8 (line 121) are cited before Figures 4 (line 169), 5 (line 173), and 6 (line 203). Line 106 says Figure 3 depicts a 5th pedal digit of a cat, but Figure 3 depicts a human distal phalanx. Further the figure captions of the figures that do depict cat digits (Figs. 4 and 5) say the imaged digits are 4th pedal digits, not 5th. Figure captions should be proofread to ensure consistency with the text.

Experimental design

The author’s materials and methods are focused on describing how imaging of specimens was performed, but it does not explain what criteria they used to differentiated between the parts of distal phalanges that are derived via endochondral ossification versus intramembranous ossification. This should be explained in the text as it leaves for some ambiguities of interpretation. For example, in the human embryo of Figure 1, a condensation of stain at the dorso-distal aspect of the distal phalanx is labeled as the bony cap. However, this condensation appears to extend far along the ventral surface of the distal phalanx and to a lesser extent along its dorsal surface. Do the authors consider this tissue to be an extension of the bony cap? Given their subsequent identification of the bony cap as restricted to the distal tip in the adult (Fig. 2), it would seem that this part is not considered an extension of the bony cap. How then, do the authors determine where the bony cap stops? Perhaps this is obvious for someone who is well-versed in histology, but an explanation here would make this paper more broadly accessible. Further, it is not obvious how the authors infer the boundary between intramembranously derived and endochondrally derived portions of the bone which they highlight in red in subsequent figures of adult distal phalanges. Is this based solely on the appearance of external morphology? I’m not sure that morphological differences between distal and more proximal portions of the distal phalanx are sufficient for such a determination. Is it not possible that endochondral and intramembranous bone in different regions of the distal phalanx may take different forms, not because of its embryological precursor, but because of interaction with other tissues? I am not arguing that the distal aspect of the distal phalanx is not formed via intramembranous ossification nor that the proximal part is not formed via endochondral ossification, but rather that gross anatomical differences alone may not be sufficient to identify the location at which one is fused to the other because gross anatomical differences may be related to more than one factor (e.g., pressure from blood vessels, ligamentous insertions, and tendinous attachments). The authors should expand upon how they have come to these conclusions in the three species they have studied.

Validity of the findings

1) The validity of findings is difficult to assess without a clear understanding of the criteria that the authors are using to distinguish the bony cap from the rest of the distal phalanx (discussed in the experimental design section of this review).
2) The authors infer that the bony cap in humans is restricted to the tuft-like expansion at the distal tip of the digit (which accounts for only a small proportion of the part of the distal phalanx that is related to the nail and its underlying dermis) and postulate that the bony cap is responsible for the formation of the nail, claw, or hoof (lines 317-332), but the authors should clarify what they mean by “formation.” It is fairly well-established in the literature that the shape of the keratinized structure and the underlying distal phalanx are related and that alterations of the bone affect the morphology of the nail. Perhaps the authors mean to say that the form of the nail follows the form of the bony cap, rather than that the bony cap is responsible for the initial formation of the nail? The tissue that generates the actual nail/claw is the proximal portion of the dermis between the nail/claw and the distal phalanx. In humans, this tissue is located proximally and does not overlay the bony cap (see Fig. 5 of Fleckman et al 2013 for its location in mice and humans), so it is unclear how the bony cap is responsible for the generation of the keratinized structure.
2.a. Lines 325-326 in the same section state that the claw of the cat forms directly distal from the bony cap. Perhaps the authors mean to say that the claw extends distally beyond the bony cap? Following Homberger et al 2009, the bulk of the keratinized components of the cat claw is generated by the dermal tissues that are located proximal and dorsal to the bony cap.
3) Regarding lines 339-340: “Further, bone regeneration at the distalmost aspect of the distal phalanx has an inductive role in bone regrowth.” What does this mean? From the cited study (Sensiate and Marques-Souza 2019), it seems that there the signals that induce appositional regrowth of a partially amputated distal phalanx originate from the nail epithelium.
4) The authors’ initial description of the proximal attachments of the “interosseous ligaments” of the human distal phalanx are incorrect (lines 157-158). These ligaments do not attach to the middle phalanges, but are rather “intraosseous” ligaments that extend from tubercles on the lateral aspects of base of the distal phalanx to spines at proximo-lateral apices of the tuft-like portion at the tip of the distal phalanx (Shrewsbury and Johnson 1975).
4.a. Later in their discussion (lines 297-303), the authors refer to these ligaments as further evidence of unique embryological development of the bony cap, but their argument as to why is difficult to follow. They should provide additional citations and better clarification on what they mean.
5) Lines 305-306 in the discussion state that the bony cap was the only ossified structure within the developing limb of their 38-day-old cat fetus shown in Figure 5, but their scout view clearly shows additional bone density structures. What then are those structures with similar density?

Additional comments

The authors identify an embryological feature and discuss its role in formation of the adult distal phalanx. They review and re-affirm the literature on humans, cats, and horses on the ossification pattern of the distal phalanx, specifically that the distal portion is derived via intramembranous ossification while the proximal portion is derived via endochondral ossification. They emphasize that the two elements are clearly distinguishable early in development, and that their adult derivatives can be identified even after the two portions fuse. The complex interplay between bone and associated keratinized appendage is an interesting topic and has relevancy for understanding the regeneration of complex structures following partial amputation. The potential of the identification of distal phalanx morphology that is indicative of a complex evolutionary history can inform and support our current understanding of horse evolution. Overall, I think this is a very interesting topic, but the authors’ anatomical descriptions are a bit ambiguous and their interpretations are sometimes difficult to follow. The suggestions I have made in this review are written with the intention of helping the authors clarify points of confusion. However, my biggest concern is that it may not be possible to infer the precise boundaries in adult distal phalanges of cats and humans based on external morphology alone as morphology is influenced by complex interplay of multiple factors, including relationships to and attachments of soft tissue. I would also like to mention that I am most familiar with claw- and nail-bearing digit tips and am much less so with hoof-bearing ones. Therefore, I have avoided directly commenting on the authors’ descriptions and interpretations of the horse distal phalanx.

Also note, that full citations of the studies referenced in this review can be found in the reference section of the reviewed manuscript with the exception of the following:
Shrewsbury M, Johnson RK. 1975. The fascia of the distal phalanx. The Journal of Bone and Joint Surgery 57:784-788.

Reviewer 2 ·

Basic reporting

A few small suggestions in comments below.

Experimental design

It is original research and advances knowledge on the topic. There are enough details to replicate the study.

Validity of the findings

Conclusions are valid. See below for further comments.

Additional comments

Smith et al. PeerJ.
The bony cap and the dual nature of the distal phalanx in humans, cats, and horses.

The authors present details of the skeletal development for the distal phalanx in fetuses and adults of three mammals. Through their histological analysis and comparisons with 19th century studies on this structure, they bring awareness that the two bone types, endochondral and intramembranous, together form the distal phalanx. Since the more proximal phalanges do not have intramembranous contributions, this dual nature of the distal phalanx may have implications for evolutionary modifications, although the mechanism of influence is unclear.

The morphological and histological detail of the distal phalanx is interesting, and this study would bring the distal phalanx into the small group of skeletal elements that are made of combination of endochondral and dermal bone. This is one topic where the authors could expand the discussion. Is there something unique about these “elements of dual nature” that could be insightful about the development and evolution of the distal phalanx?

The uniqueness of the distal phalanx vs the proximal phalanges is recognized in several developmental studies not cited.
e.g. Casanova et al., 2007. Digit morphogenesis: Is the tip different? Development, Growth, and Differentiation. https://doi.org/10.1111/j.1440-169X.2007.00951.x
Also, the distal phalanx in birds is regulated as a distinct module in development (shown experimentally in Kavanagh et al., 2013 PNAS).

Is the bony cap specific to mammals?

Figure
“The bony cap (red) at this stage is entirely independent of the distal phalanx proper.” It doesn’t look independent, but is surrounding the distal phalanx. Rephrase so it makes sense.

Line 258 – problematic grammar.

Line 265 Instead of “The bony cap has a distinct embryological origin from the cartilaginous distal phalanx” (which is the opposite of what you mean.)
Try “The bony cap has an embryological origin distinct from the cartilaginous distal phalanx”

Line 273 “certain genes..” why not say which genes?

Line 279 “is due to its growth being restricted by the ossified bony cap” seems speculative without experimental evidence of cause-effect. Rephrase to: early descriptions speculate that growth is restricted…

Line 288-293 Again, I think this idea that the unique shape of the distal phalanx is *caused * by the presence of the bony cap is interesting and possibly correct, but no one has done the experiment. However it is very intriguing and you should aim to set up the logical and comparative evidence as basis for the experimental embryological work. There are situations where surrounding embryonic bone interacts with adjacent forming organs in a complex back and forth signaling and with potential constraint to the adjacent tissue (e.g. Renvoise, et al., 2017 PNAS (tooth and jawbone); also Zhu et al. 2017, Jan. 5. Sci Rep).

Perhaps the authors could make a simple illustration of the conserved developmental origin of the bony cap separate from the distal endochondral ossification, showing that bony cap intramembranous ossification is first, followed by endochondral? Then illustrate how it diverges later in development in different species (fusions, shape change, etc.). Something like that would make the story more clear.

Line 316 and entire paragraph. Postulate that the ‘bony cap is responsible for nail formation’…
Do the authors believe that intramembranous bone (the bony cap) could house stem cells, regarding the links with regeneration? This should be clarified.

Figures– the authors should note each case when it is an adult specimen, to clarify from the fetuses.

---

## Round 0.2 · Minor Revisions

Thanks to the authors for addressing the major reviewer comments from the first round. However, as per the review below a number of smaller corrections and clarifications remain and should be addressed.

Reviewer 1 ·

Basic reporting

1) The authors have clarified in their response that they are the ones positing that the intramembranous bony cap of the distal phalanx is dermal in origin and have included some text within the MS to that affect. However, this point is confused by statements like that on lines 71-73 “Developmentally, they differ for the other more proximal phalanges and include a cartilaginous intramembranous proper section of mesodermal tissues and an intramembranous ectodermal component (Dixey, 1881; Baran & Juhlin, 1986).” The authors should do a careful proofread to ensure that text throughout is consistent and that their logic and reasoning is clearly explained to the reader.

2/3/4) For the most part, the authors have clarified and correct their captions, However, the wording in the caption for figure 4 is still a bit ambiguous. The caption says “The bony cap is outlined in red and is deep to the claw (keratin), wedged into the L-shaped distal phalanx.” It sounds like this is saying that the bony cap is wedged into the L-shaped distal phalanx, but since the bony cap is part of the distal phalanx this doesn’t make sense. Perhaps the authors mean that the claw is wedged into the distal phalanx? But that doesn’t make sense either because it is more like the ungual process of the distal phalanx is wedged into the claw… The authors have also not explained what they mean by L-shaped. I’ve looked at a lot of cat distal phalanges, but am failing to see the resemblance.

Experimental design

The authors have adequately addressed my concerns here. :)

Validity of the findings

The authors have adequately addressed most comments here, but I am confused by their text on lines 228-229: “We propose there are likely interosseous ligaments that create two tunnels between the ligaments themselves and this distal phalanx.” Assuming that they are still talking about the intraosseous ligaments from the preceding sentence – this has already been well-established by Shrewsbury and Johnson. If my assumption is incorrect and the authors are discussing a different set of ligaments, they should clarify what ligaments they are talking about.

Additional comments

While I feel that this manuscript has been greatly improved and clarified by the new additions, I think the authors need to do another careful proofreading of their text. For example, in newly added lines 55-56 “present for bone distal phalanx bone regrowth” should probably be “present for distal phalanx bone regrowth,” line 57 “distal zones are lesser understood than appreciated” should be “less understood,” and line 58 uses the word mesenchymenly which I don’t believe exists. I’ve noticed a number of little such typos throughout the text.

---

## Round 0.3 · Minor Revisions

The reviewer's comments make a good case that their previous comments were not sufficiently addressed in this revision. I'll add the following additional comments, none of which rise above minor issues. But together they support the conclusion that the manuscript, albeit being close, isn't fully ready yet for publication.

L60-62: "also" occurs redundantly in the clause starting with "but transplant experiments"
L64: "this intramembraneous region between the two remains to be determined" - determined as what? Perhaps the authors mean to say "the origin of this intramembraneous region ..."?
L92: Due to the beginning of the sentence ("In our study") it's unclear whether "is named the bony cap" refers to a deliberate choice made by the authors, or to how the structure is generally named in morphology. If the former, consider using active voice ("we name ..."), and if the latter, please change sentence to make this clear.
L119: "for this study" is duplicated
L301: for consistency with other locations in the text, shouldn't "digit three" be written as "digit III"?
L317: "but have not identified" should be "but has not identified"
L318: "The bony cap was identified in humans, cats, and horses." In this particular study, or in general (meaning, not yet in other taxa)? Either way, please make clear.
L324: "The bony cap is observed in all digits, pes and manus alike." In this study, or is this a fact that would need a citation?
L340-341: "The complex nature [...] is misunderstood, incompletely described, or sometimes simply ignored." Because there is a qualifier only on the "simply ignored", as written this implies that the complex nature is always or almost always misunderstood or incompletely described. Do the authors intend to make a claim quite as strong? If not, consider saying e.g. "frequently misunderstood".
L344: "due to its growth is restricted": "is" should be "being"
L351: "which they had termed the unguicular process": remove "had"
L364-365: "the distal phalanx has a dual nature of embryological origin" The following sentence says this is a speculation, which means this should be stated as one, not as a fact. E.g., "Thus, we posit that ..."
L391: "the bony cap becomes filled with the distal phalanx and will fuse into one entity": Need multiple entities to fuse, e.g. "... and both will fuse into one entity"
L401-402: "The bony cap ..." This sentence as written seems to lack a verb in the main clause.

Reviewer 1 ·

Basic reporting

1) I do not believe that my first Basic Reporting comment from the second review was adequately addressed. The original comment is as follows: “The authors have clarified in their response that they are the ones positing that the intramembranous bony cap of the distal phalanx is dermal in origin and have included some text within the MS to that affect. However, this point is confused by statements like that on lines 71-73 “Developmentally, they differ for the other more proximal phalanges and include a cartilaginous intramembranous proper section of mesodermal tissues and an intramembranous ectodermal component (Dixey, 1881; Baran & Juhlin, 1986).” The authors should do a careful proofread to ensure that text throughout is consistent and that their logic and reasoning is clearly explained to the reader.”

For the sake of clarity, I want to note that when I am referring to line numbers, I am using those in the supplied pdf version of the manuscript as opposed to the MS word document. There seems to be a glitch causing a skip in the line numbering of the word document after line 78 so that the line numbers do not match up.

In their response, the authors stated that they have “re-read the manuscript to ensure clarity with our argument throughout the text.” Following the track changes in the supplied word document, I see that they have edited lines 71-73 to read as “Developmentally, they differ from the other more proximal phalanges such that the distal region includes both mesodermal tissues of the distal phalanx and an intramembranous ectodermal component that interacts with the phalanx (Dixey, 1881; Baran & Juhlin, 1986).” This statement still suggests that an ectodermal origin for the intramembranous component is an established fact rather than an idea posited by the authors. Lines 329-331 imply the same: “There is a distinct separation of the two bones embryologically the bony cap has a distinct ectodermal origin rather than the mesenchymal phalanx bone.”

Further, the authors have not clarified their reasoning regarding why they posit that the bony cap is ectodermal in origin. It seems that the authors might be implying that an ectodermal origin of the bony cap follows from the presence of osteogenic signals that originate from ectodermal tissues. I do not see how this follows unless the authors are erroneously equating the location of origin of osteogenic signals with the origin of the actual tissue that is responding to said signals. For example, the apical ectodermal ridge (itself derived from ectodermal cells in response to epithelial-mesenchymal interactions; reviewed in Casanova and Sanz-Ezquerro 2007) helps to guide limb growth via cellular signaling, but the mesenchymal limb tissues that it guides are not suddenly referred to as ectodermal simply because some signals that affect them are. I’m not sure that this is what the authors have intended to imply and perhaps I am missing something, but I feel this should be addressed.

Ultimately, the authors really need to 1) explain and adequately support why they think the intramembranous region of the distal phalanx is ectodermal in origin rather than mesenchymal, and 2) carefully proofread their text to ensure they are not stating their hypotheses as established facts. To address these points, the authors need to include text along the lines of, “we propose that the bony cap may be ectodermal in origin as opposed to mesenchymal because …” and provide their reasoning. When referring to any idea that is unconfirmed, it should always be described using words that denote uncertainty, such as “may be” and “might have” rather than “is” and “has.” As their introduction reads right now, an ectodermal origin of the bony cap is treated as an established fact that is attributed to papers that have not actually made this claim (Dixey, 1881; Baran & Juhlin, 1986). Even unintendedly so, that is misleading.
* * *
2) The authors have addressed my second Basic Reporting comment from the second review, but I am still confused as to what they mean. For context, the text of my earlier comment is, “For the most part, the authors have clarified and correct their captions, However, the wording in the caption for figure 4 is still a bit ambiguous. The caption says “The bony cap is outlined in red and is deep to the claw (keratin), wedged into the L-shaped distal phalanx.” It sounds like this is saying that the bony cap is wedged into the L-shaped distal phalanx, but since the bony cap is part of the distal phalanx this doesn’t make sense. Perhaps the authors mean that the claw is wedged into the distal phalanx? But that doesn’t make sense either because it is more like the ungual process of the distal phalanx is wedged into the claw… The authors have also not explained what they mean by L-shaped. I’ve looked at a lot of cat distal phalanges, but am failing to see the resemblance.”

The authors have edited their Figure 4 caption to state “The distal phalanx folds over the bony cap and the bony cap is wedged into the claw,” but I am still struggling to visualize what "the distal phalanx folds over the bony cap" means. I think it would be helpful if the authors were to describe the morphology using terms of directionality which would help clarify the ambiguities raised by their current phrasing. While I understand what is meant by the bony cap being wedged into the claw, it could be made even more clear by saying something like “the keratinous sheath folds over the dorsum of and covers the left and right sides of the bony cap.” The latter leaves much less to interpretation than the former because there are a number of conceivable ways in which one structure can be wedged into another (the sheath covers the dorsum, ventral surface, left and right sides of the ungual process; the sheath covers the ventral, left and right sides of the ungual process; etc.). I don’t have a specific suggestion as to how to describe the relationship of the endochondral portion of the distal phalanx to the bony cap because I don’t understand which portions are considered to be which. However, I suggest using an anatomical description similar to what I suggested for the relationship between the claw and bony cap in which specific sides (i.e., proximal, volar, left, right, distal, etc.) are referenced.

Not only is the current wording too open to interpretation, but there are inconsistencies among figures, in the terminology used to denote different parts of the distal phalanx, and between the text of the results and discussion sections that are obfuscating the authors' description. I’ve discussed each below:
- Regarding the figures, the portion that is denoted as the bony cap is inconsistent between Figure 4 and Figure 5; at least a portion of the proximal articular facet is highlighted in red in Figure 4 (and thus indicated as part of the bony cap), but it is not at all highlighted in Figure 5 (and thus indicated that it is not part of the bony cap). The text of the paper as well as Homberger et al.’s (2009) findings imply that the proximal articular facet is part of the endochondral portion of the distal phalanx, and therefore Figure 4 is in error.
- Regarding the inconsistent use of terminology for parts of the distal phalanx, the figure caption for Figure 4 describes the distal phalanx folding over the bony cap, and so it would seem that in the context of this figure caption, distal phalanx is being used as synonymous with the endochondral portion of the distal phalanx. However, in the introduction, the bony cap is defined as part of the distal phalanx; line 54: “The exception to this is the apex of the distal phalanges, the bony cap,” and lines 78-79 imply that both portions are considered to be the distal phalanx: “Consequently, the distal phalanx has a dual nature: the intramembranous bony cap and the remaining endochondral distal phalanx.” Confusion regarding to how the authors are using the phrase distal phalanx in different contexts is likely exacerbating my confusion in the interpretation of their anatomical descriptions and partly explain the inconsistencies between the results and discussion sections.
- Regarding inconsistencies between the results and the discussion sections, lines 221-224 state: “The unguicular hood is labeled in the literature, but due to our determination of the bony cap we believe the unguicular hood is the distal phalanx. We deduced that the distal phalanx is a specialized bone with a curved “L” shape and a deep median groove (Fig. 4). Inside that groove, the thick hollow proximal end of the bony cap fits congruently and is fused in the adult (Fig 5a).” I also see that in Figure 4, the flexor tubercle is labeled as the distal phalanx. I interpret this to mean that the endochondral portion of the distal phalanx is restricted to the unguicular hood, the flexor tubercle, and the proximal articular facet, while the bony cap would be the ungual process (or remainder of the distal phalanx). I interpret the deep median groove to be the space bracketed by the unguicular hood on the left and right sides, the flexor tubercle on the volar side, and the proximal articular facet on the proximal side. However, the implied inclusion of the unguicular hood as part of the endochondral portion of the distal phalanx is contradicted by lines 334-336 in the discussion [“Homberger et al. (2009) does indicate that the unguicular hood (which extends from the endochondrally derived portion of the distal phalanx) is ossified intramembranous several months after birth”]. Rather it seems that the unguicular hood is a third, distinct part of the feline distal phalanx that ossifies in its own distinct manner and is neither the bony cap nor the endochondral portion.

In summary, the authors should clarify what they mean using less ambiguous anatomical terminology/directional terms and ensure that their figures (especially regarding the proximal articular facet in Figures 4 and 5) and text support a clear and consistent argument. The authors should not use the phrase “distal phalanx” to refer to only the endochondral portion of the distal phalanx without denoting that they are doing so and without maintaining consistency in usage throughout the paper. However, I strongly advise against redefining “distal phalanx” to refer only to the portion of that bone which develops endochondrally. Doing so would inject unnecessary confusion into discussions of distal phalanx anatomy because it would contradict the well-established terminological usage of the phrase “distal phalanx.” Even other researchers who have studied the intramembranous ossification of its tip have continued to use the phrase to refer to both portions.

Experimental design

n/a :)

Validity of the findings

My previous comment was addressed. :)

Additional comments

The authors have mostly addressed my comment from the second review: “While I feel that this manuscript has been greatly improved and clarified by the new additions, I think the authors need to do another careful proofreading of their text. For example, in newly added lines 55-56 “present for bone distal phalanx bone regrowth” should probably be “present for distal phalanx bone regrowth,” line 57 “distal zones are lesser understood than appreciated” should be “less understood,” and line 58 uses the word mesenchymenly which I don’t believe exists. I’ve noticed a number of little such typos throughout the text.”

There are still some minor typos that the authors have not caught, and this includes one that I noted in my previous comment (lines 55-56 “present for bone distal phalanx bone regrowth”).
* * *
Also, I would like to acknowledge the wordiness of my comments in this round of the review. I suspect that my shorter comments from the second review weren't as clear as they could have been and perhaps undercut the importance of some of the points I was trying to make. Therefore, I tried to explain in more detail and be more explicit in this round. While it may seem like I am suggesting a lot, I do not think these constitute major revisions as most of my requests are geared towards clarifying ambiguities and inconsistent phrasing in the text.

---

## Round 0.4 · Minor Revisions

I appreciate the improvements, but upon a careful read through myself there continue to be a considerable number of issues. Many of these are minor typographical, grammatical, and punctuation errors (which could perhaps be passed on to the galley proof stage), but some remain that, even if minor in nature, require a decision and changes by the authors as to how to revise the text to address them. To expedite this process I am attaching an annotated version (through tracked changes and comments) based off of the clean (not change-tracked) Word document. I hope this helps resolving the remaining issues with less confusion about line numbers (which unfortunately seem to differ between each of the PDF, Word clean, and Word change-tracked versions). Please note that there are corrections and revision-requiring comments for the figure captions too (at the very end of the document).

---

## Round 0.5 · accepted · Accept

Thanks for the attention to detail in resolving the remaining comments. I agree the manuscript is now ready for publication.